# Process modelling of NHS cardiovascular waiting lists in response to the COVID-19 pandemic

Salvador Catsis,[1] Alan R Champneys,[1] Rebecca Hoyle [ORCID],[2] Christine Currie,[2] Jessica Enright,[3] Katherine Cheema,[4] Mike Woodall,[5] Gianni Angelini [ORCID],[6] Ramesh Nadarajah [ORCID],[7,8] Chris Gale,[7,8] Ben Gibbison [ORCID][9,10]

For numbered affiliations see end of article.

**Correspondence to**
Dr Ben Gibbison;
ben.gibbison@bristol.ac.uk

## ABSTRACT

**Objective** To model the referral, diagnostic and treatment pathway for cardiovascular disease (CVD) in the English National Health Service (NHS) to provide commissioners and managers with a methodology to optimise patient flow and reduce waiting lists.

**Study design** A systems dynamics approach modelling the CVD healthcare system in England. The model is designed to capture current and predict future states of waiting lists.

**Setting** Routinely collected, publicly available data streams of primary and secondary care, sourced from NHS Digital, NHS England, the Office of National Statistics and StatsWales.

**Data collection and extraction methods** The data used to train and validate the model were routinely collected and publicly available data. It was extracted and implemented in the model using the PySD package in python.

**Results** NHS cardiovascular waiting lists in England have increased by over 40% compared with pre-COVID-19 levels. The rise in waiting lists was primarily due to restrictions in referrals from primary care, creating a bottleneck postpandemic. Predictive models show increasing point capacities within the system may paradoxically worsen downstream flow. While there is no simple rate-limiting step, the intervention that would most improve patient flow would be to increase consultant outpatient appointments.

**Conclusions** The increase in NHS CVD waiting lists in England can be captured using a systems dynamics approach, as can the future state of waiting lists in the presence of further shocks/interventions. It is important for those planning services to use such a systems-oriented approach because the feed-forward and feedback nature of patient flow through referral, diagnostics and treatment leads to counterintuitive effects of interventions designed to reduce waiting lists.

## STRENGTHS AND LIMITATIONS OF THIS STUDY

⇒ Systems dynamics modelling provides and overview of the whole healthcare system rather than treating each waiting list as a separate entity.
⇒ The systems dynamics model provides the opportunity for policy-makers to test interventions that change patient flow (eg, increasing capacity of consultation, diagnostics or treatment) and estimate their effect on patient flow.
⇒ The model uses publicly available, routinely collected time-series data.
⇒ The model only uses aggregated national data, and therefore, cannot be applied at the level of a particular region or hospital.
⇒ The model does not account for the changes in patient health that occur as a result of the changes in waiting list—some interventions may be more impactful at improving health than others.

CVDs are progressive, and therefore, timely treatment prevents both death and serious morbidity (with a subsequent impact on healthcare costs). Care of CVD comprises a multidisciplinary pathway involving primary, secondary and tertiary care. Saturation of capacity at any point can lead to downstream bottlenecks, manifesting as increased waiting lists of referral, diagnostics or treatment (RDT). The RDT pathway has a number of feedback loops that occur in patient flow (see figure 1), which means that the effect of point changes can be neither obvious nor intuitive. Therefore, a systems-level approach must be taken to maximise patient flow, timely patient care and improved clinical outcomes.

Data from January 2021[3] showed over 230 000 people waiting for invasive heart procedures and heart operations in England (with around 4500 waiting over a year, over 150 times the equivalent figure before the pandemic). These numbers increased further in the subsequent year, and the impact of COVID-19 on waiting times will persist beyond

## INTRODUCTION

Cardiovascular disease (CVD) affects over 7 million people and accounts for 27% of all deaths in the UK.[1] Since the start of the COVID-19 pandemic in March 2020, CVD continues to be the largest non-COVID-19 cause of excess mortality in England.[2] Most

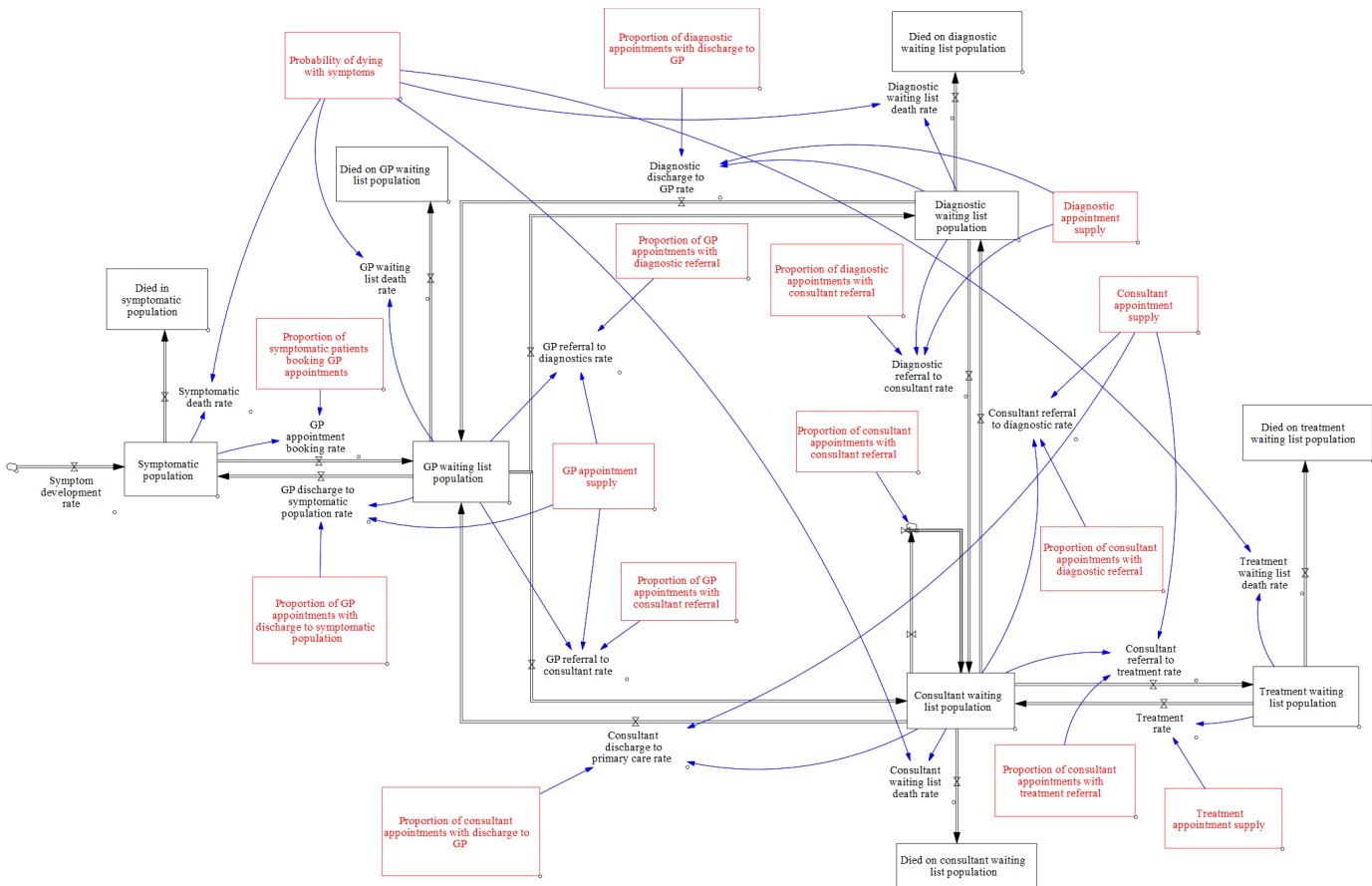

**Figure 1** Overview of the process model. Symptomatic patients S enter from the left in the diagram. There are four other waiting-list stocks, a primary care general practitioner (GP) waiting list G, a secondary care diagnostic waiting list D, a consultant waiting list C and a secondary care treatment waiting list T. Each stock is represented by a black box in figure 1 arrowed pipes represent flows of patients. α and β parameters are represented by red boxes in figure 1, and the flows they influence are represented by blue arrows.

the point at which it becomes endemic. The aim of this study was to model National Health Service (NHS) waiting lists in England for adult CV care to provide insights and potential solutions as to the optimal approaches to maximise patient flow and improve patient care. We adopted a systems dynamics modelling approach[4] and built a data-informed model that can predict adult patient flow over time through CV treatment pathways.

### METHODS
#### Overview of model

An overview of the mathematical model is given in figure 1. The model assumes that patients are stocks in a process flow diagram (represented by a black box in figure 1) in which symptomatic patients S enter from the left in the diagram. There are four other waiting-list stocks, a general practitioner (GP) waiting list G, a secondary care diagnostic waiting list D, a consultant waiting list C and a secondary care treatment waiting list T. Only non-emergency admissions are captured, although the effect of emergency admissions might be reflected in the data on which we fit parameters, for example, in the availability of non-emergency treatments. Stocks are

included for those that leave the system during each stage of the process. For simplicity, we defined these stocks as representing those who have died, but in reality people may leave the system for other reasons such as movement to a private healthcare system.

The arrowed pipes in the diagram represent flows of patients. Flow rates are numbers of patients from the outgoing stock that transition to the receiving stock per unit of time, defined as the product of two parameters. First, is a capacity parameter $\alpha_i$ which is the overall rate at which stock i is serviced, where $i \in \{S, G, D, C, T\}$. This is multiplied by a decision parameter $\beta_{ij}$ which is the proportion of stock i that flows to stock j at that time step. Most of the α and β parameters are represented by red boxes in figure 1, and the flows they influence are represented by blue arrows.

The figure depicts multiple pathways. It is assumed that symptomatic patients in the community who do not die, eventually book an appointment with their GP. After GP consultation, there can be a number of outcomes: (1) discharge to the symptomatic population, (2) diagnostic tests or (3) direct referral to a specialist (consultant) without diagnostic tests first. After diagnostic tests,

patients can return to primary care (the GP) or continue in secondary care and see a consultant. The consultant can then (1) discharge the patient back to their GP, (2) send them for further diagnostic tests, (3) send them to a treatment pathway (eg, surgery, interventional cardiology) or (4) return for follow-up. Thus, the model has several possible loops, which can give rise to feedbacks and non-intuitive behaviours.

Unlike in infectious disease modelling, there is no stock that represents recovery or cure; instead the successfully treated are returned to their GP and ultimately go back to the symptomatic population. There are several justifications for this simplification. First, the model does not track individual patients. Instead, we include a parameter $\beta_S$ which is a top-up rate of symptomatic patients, that represents the balance in the general population between those developing new symptoms and those either whose symptoms have been resolved with treatment or have died. To capture mortality, a 'hidden' stock M is included in the model, with mortality rate from each stock equal to $\alpha_i \beta_{iM}$. Under this definition $\sum_j \beta_{ij} = 1$, where $I, j \in \{S, G, D, C, T, M\}$. Second, the nature of the conditions we are modelling tend to be progressive, and patients typically require continued follow-up—although the frequency of consultations may reduce. Finally, we consolidated both medical and surgical pathways into a single patient stream. Within the conditions we have grouped, there will not only be heterogeneous rates of recovery, but differing diagnosis, treatment and consultant appointment constraints.

The model was implemented using the PySD package in python.[5] The rate of change of each stock S, G, D, C and T is set equal to the sum of all incoming flows minus outgoing flows. This results in a system of five ordinary differential equations. The historical data is used to fit the $\alpha_i$ and $\beta_{ij}$ parameters, so that to make a model run we need initial values for each of the five stocks and values for $\alpha$ and $\beta$ parameters (which may be time varying). Hence the modelling is a two-stage process. First, we make parameter estimates based on historical data, known future trends

or on hypothetical interventions in the form of changes in capacity constraints (the $\alpha$ parameters) or decision variables (the $\beta$ parameters). Then the model is run to produce future predictions of waiting lists.

## Data sources

Table 1 summarises the routinely collected publicly available data streams used, which we sourced from NHS Digital (NHSD), NHS England, the Office of National Statistics (ONS) and StatsWales. StatsWales data was used to remove Wales' mortality data from the 'England and Wales' mortality data to leave mortality data for England only. Further details of the precise data streams we used and how we processed them to enable plotting over comparable time intervals is given in online supplemental materials.

Data were sliced to capture a set of CV conditions whose prevalence rate data were routinely collected at the primary care level and published in the Quality and Outcomes Framework.[6] The conditions captured are: atrial fibrillation, coronary heart disease, heart failure (HF) and hypertension (HYP). We chose to omit CVD primary prevention, as these data are not reported from 2020 onwards.

Monthly diagnostic and hospital episode statistics (HESs)[7] are coded differently. We included data and hospital episode data that are routinely collected and published in the monthly diagnostics data and the HES monthly series, respectively. The chosen treatment function codes for the HES monthly series are: cardiothoracic surgery service (170), cardiac surgery service (172), cardiothoracic transplantation service (174), cardiology service (320), cardiac rehabilitation service (328), transient ischaemic attack service (329) and congenital heart disease service (331). For the diagnostic data, we chose 4 of the 10 diagnostic treatment codes: echocardiogram, electrophysiology, CT scan and MRI scan. For the latter two, we used historical data available from one NHS commissioning support unit[8] to estimate the proportion

| Table 1 | Sources of routinely collected data used in this paper | | |
|---|---|---|---|
| **Data series name** | | **Source** | **Time step of data update** |
| Quality and outcomes framework[6] | | NHSD | Annually |
| Appointments in general practice[18] | | NHSD | Daily |
| Monthly diagnostics data[19] | | NHSE | Monthly |
| Consultant-led referral to treatment[9] | | NHSE | Monthly |
| Patients registered at a GP practice[20] | | StatsWales | Annually/quarterly |
| Deaths registered in England and Wales[21] | | ONS | Annually |
| NHS sickness absence rates[22] | | NHSD | Monthly |
| Hospital episode statistics—admitted and outpatient[7] | | NHSD | Monthly |

Deaths registered in England and Wales by the ONS[19] are each given a unique International Classification of Diseases 10th Revision code. We used data with the following codes; I00–I02 acute rheumatic fever, I05–I09, chronic rheumatic heart disease, I10–I16 hypertensive diseases, I20–I25 ischaemic heart diseases, I26–I28 pulmonary heart disease and diseases of pulmonary circulation, and I30–I50 other forms of heart disease.
GP, general practitioner; NHSD, National Health Service Digital; NHSE, NHS England; ONS, Office of National Statistics.

of those tests that were for the six conditions captured in our analysis. The consultant-led referral to treatment (RTT)[9] data, which records time from referral through to treatment in the planned treatment pathway. Here data are only recorded for the 18 leading treatment functions in the quality and outcome framework, with other conditions being given a general X coding. We use data from two of these 18 codes, cardiology services (170) and CV surgery services (320).

## Data processing and parameter fitting

All data are interpolated to be represented as figures per typical day, defined as one seventh of a week, for which the effects of weekends and public holidays are smoothed out. Each time series is divided into three epochs; prior to the initial 2020 restrictions in England, during those restrictions (taken as 26 March 2020–15 June 2020) during which patients were discouraged from accessing healthcare services except in emergencies, and the period from 16 July 2020 to 31 January 2022 in which a range of partial COVID-19 restrictions were in place. See online supplemental materials for details of the data processing used.

To fit parameters to the model, we assume that data in the first epoch was in steady state. Data from this epoch are then used to set the initial conditions for all the stocks in the model and to estimate all of the $\alpha$ and $\beta$ parameters, as we briefly describe here (see online supplemental materials for details). All of the $\alpha$ parameters, governing appointment capacities, and the $\beta_{GC}$ and $\beta_{CG}$ parameters, governing referrals to and from GPs to consultants can be directly estimated from the data. Other parameters can be directly inferred using a steady-state assumption. There remain a few parameters that cannot be inferred. First, the mortality rate at each stage on the waiting list is uncertain. Instead, we used the simplifying assumption that the mortality rate $\beta_{iM}$ is the same for all symptomatic patients, regardless of where they are in the system. This rate is readily calculated from the data, and changes little during the study period. Second, the net rate $\beta_S$ of new symptomatic patients is not readily available, nor is the proportion $\beta_{CC}$ of consultant appointments that result in a further appointment rather than treatment or diagnosis. We use simple least-squares optimisation to estimate values for these two parameters.

During the second and third epochs, we assumed continuity of stocks from the end of the previous epoch, but repeat a fitting procedure for the parameters, under the assumption that the system was no longer in steady state. Again, all the $\alpha$ parameters and $\beta_{CG}$, $\beta_{GC}$ can be directly estimated from the data, and the mortality parameters are set to be the same as in the previous epoch. We did not use a steady-state assumption for the other parameters, but fixed their values using least squares optimisation to the recorded time-series data during that epoch.

Finally, to make predictions of future behaviour of waiting lists, the model is run into a fourth epoch, from February 2022 onward. Model parameters that were fit

during the third epoch are used. To introduce an element of uncertainty into the predictions, we used a simple Monte Carlo simulation method, by adding 10% uniform random variation to the fitting parameter $\beta_{CC}$ (and consequent proportional adjustments to $\beta_{Cj}$ parameters, for $j \neq C$, to ensure the constraint $\Sigma_j \beta_{ij}=1$ is maintained). To capture the effect of simple future interventions designed to reduce waiting lists, we consider changes to the capacity parameters $\alpha_i$.

## Patient and public involvement

There was no patient or public involvement in this study.

## RESULTS

### Description of the data

Figure 2 shows the state of the waiting lists for cardiac surgery and cardiology treatment, respectively. Both have risen by about 40% from their prepandemic levels. There was no significant rise in these waiting lists during the first restriction period. However, a number of new RTT pathways (both cardiology and cardiac surgery) were halved during the initial restrictions (see online supplemental materials); this significant demand reduction, coupled with some elective work continuing, resulted in an initial stabilisation of the waiting list. Indeed, figure 2B shows the cardiology waiting list fell during the initial restrictions, whereas figure 2A shows the surgery waiting list remained static. The rise in waiting lists for treatment after the first set of restrictions in 2020 was primarily due to restrictions in primary care reducing the supply of patients to secondary care, which created a 'bottleneck' after this period had stopped. For example, figure 2C shows an almost twofold reduction in patients seeing their GP for CV coded conditions during lockdown. These numbers rose almost immediately after the first period of restrictions to a mean slightly above their prepandemic levels, although with somewhat greater fluctuation.

Another effect of reduced hospital capacity can be seen in the data for the number of patients having echocardiograms (figure 2D), which fell significantly during the first period of COVID-19 restrictions and were slow to return to prepandemic levels. Waiting lists for such tests continue to rise (figure 2E). Data for other diagnostic procedures show similar trends (see online supplemental materials). Thus, it would seem that the rise in secondary-care waiting lists is in part due to the only gradual return of diagnostics (and presumably other ancillary services) to their prelockdown capacities.

### Predictive model results

The results of the model simulations are presented in figure 3. They illustrate both the fit to the historic waiting list data and the predictions under different interventions. Further graphs are presented in online supplemental materials.

Figure 3A shows the waiting list for treatment within the model, under the assumptions of no intervention, or

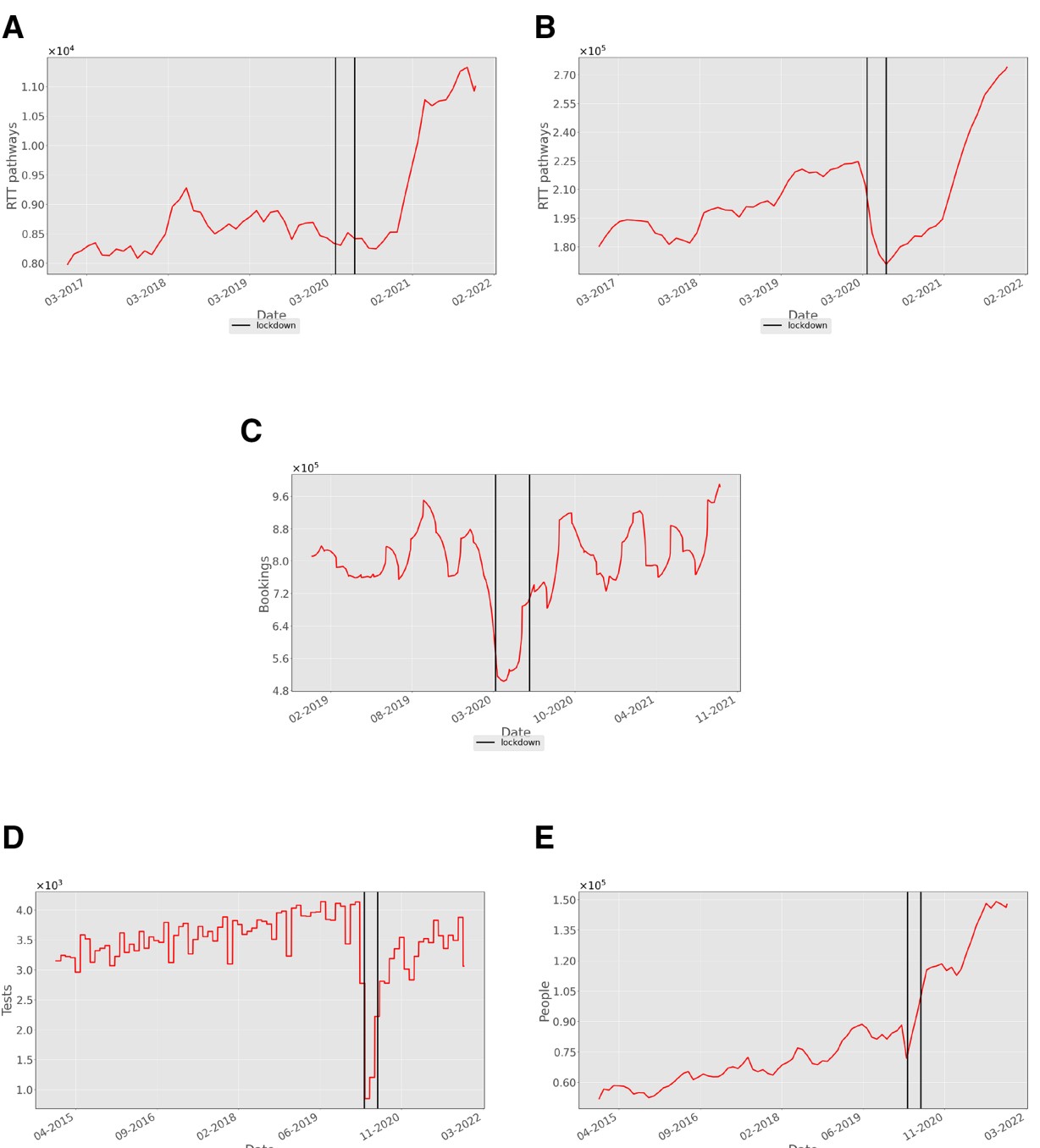

**Figure 2** Numbers of (A) cardiothoracic surgery and (B) cardiology patients waiting to start NHS treatment in England, sampled monthly from January 2018 to December 2021. (C) Number of daily GP booked appointments (×100 000) for cardiac conditions reported by NHS England over a time period from 2019 to 2021. (D) Number of NHS echocardiogram tests (×1000) carried out per day in England from 2015 to 2021. (E) The consequent size of the waiting list for echocardiograms (×100 000). The three epochs in the data are separated by vertical black lines, so that the thin region between the two vertical lines represents the period of initial lockdown in England from 26 March 2020 to 15 June 2020. GP, general practitioner; NHS, National Health Service.

an increase of either 5% or 10% in treatment capacity. Taken alone, this graph shows that a 10% increase in the supply of treatment would lead to the waiting list being cleared within 6 months. However, figure 3B shows that simply treating more patients would not reduce the waiting list for consultant appointments, which is an order of magnitude larger (about 300 000 compared

with 30 000 in February 2022) and would continue to rise. Moreover, the intervention of increasing treatment capacity causes the initial consultant waiting list to rise, presumably because those undergoing the additional treatments are referred back to consultants before discharge to primary care. This intervention also does not improve diagnostic or GP waiting lists (see online

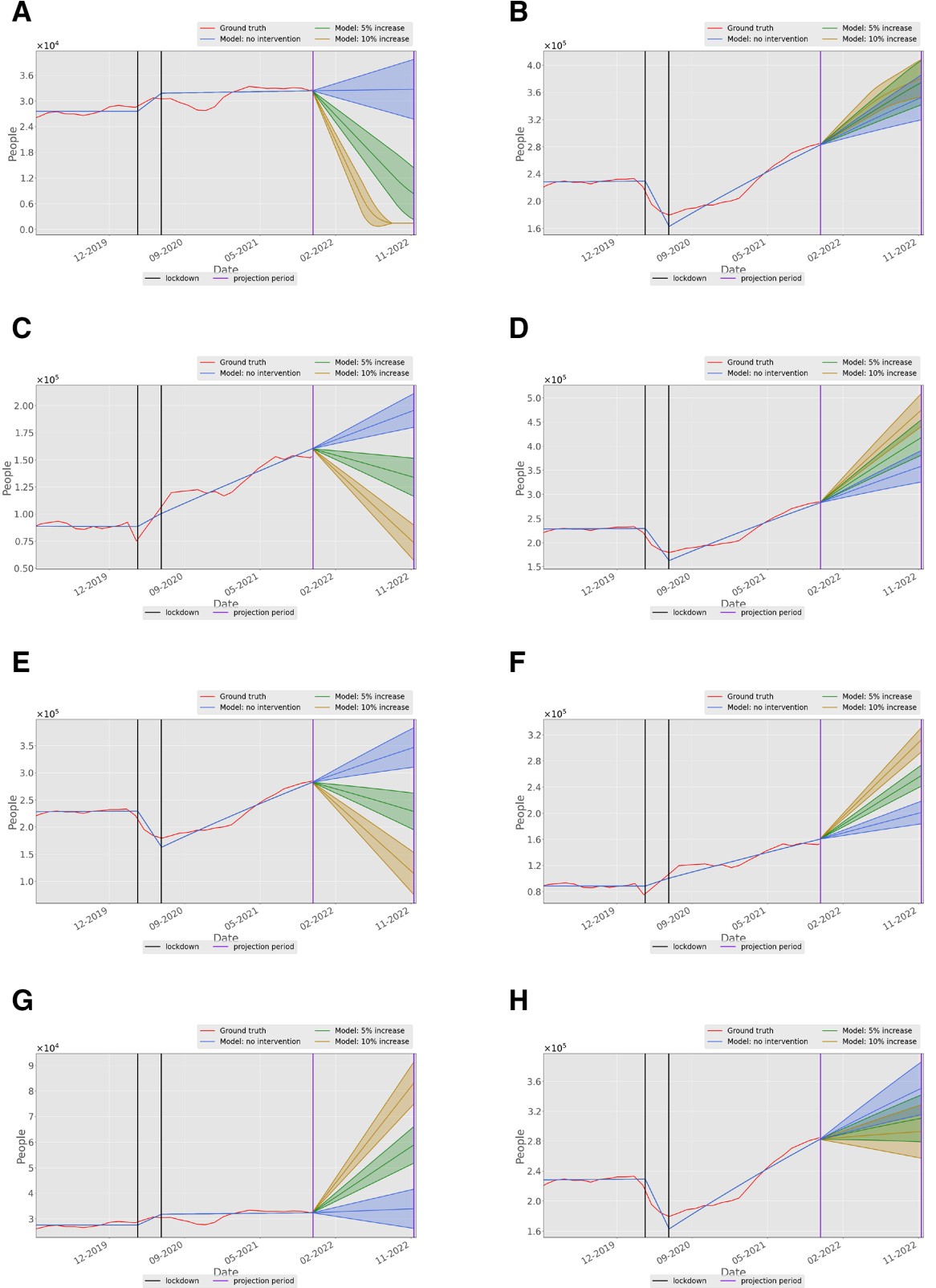

**Figure 3** Model output for the (A) treatment waiting list and (B) consultant waiting lists populations, showing the projected effect of an increase in the treatment appointment supply on the size of the treatment waiting list. (C) Model output for the diagnostic and (D) consultant waiting list population, showing the effect an increase in the diagnostic appointment supply. (E) Model output for the consultant waiting list population, showing the effect of an increase in the consultant appointment supply. (F) Model output for the diagnostic and (G) treatment waiting lists, showing the effect of an increase in the consultant appointment supply. (H) Model output showing the effect of increasing the appointment capacities of the GP, consultant, diagnostics and treatment simultaneously. In this figure, shading around the central prediction shows the envelope of outcomes that result from including a 10% uncertainty factor in the model fitting parameter. GP, general practitioner.

supplemental materials) although the effect is less dramatic.

Figure 3C shows the effect of increase in supply of diagnostics. Again, a 10% increase in supply can rapidly reduce the waiting list—reduced from 150 000 to 5000 within 9 months. However, figure 3D shows that this intervention is also counterproductive, as the consultant waiting list increases by almost 200 000 over the same period, rising at a far steeper rate than it would without the increase in diagnostic appointments. There is little effect on GP or treatment waiting lists (see online supplemental materials).

The consultant appointment supply is the largest determinant of flow within the model. Figure 3E shows that a 10% increase in the supply of consultant appointments has the effect of approximately halving the consultant waiting list within 9 months, compared with a near doubling without intervention. Yet, as shown in figure 3F,G this increase in consultant appointment supply without increase in diagnostics or treatment has an indirect effect on diagnostic and treatment waiting lists, both of which rise as a consequence. However, the net effect compared with the 'do-nothing' scenario is an increase of about 60 000 in treatment waiting lists and 100 000 in diagnostic waiting lists over 9 months compared with a reduction of about 250 000 in the consultant waiting list. So, the net effect of increasing the consultant appointment supply would be to reduce the overall number of waiting patients.

Finally, figure 3H shows the effect of increasing the appointment capacities of the GP, consultant, diagnostics and treatment simultaneously. Intuitively this would lead to a bigger improvement. Yet, paradoxically, the model shows that this combined intervention is less effective at reducing consultant waiting lists than a similar-sized intervention to the consultant waiting list alone. In fact, this combined intervention also has a weaker effect on the diagnostics and treatment waiting lists than a similar-sized intervention on those waiting lists alone (see online supplemental materials[10]).

## Discussion

The COVID-19 pandemic has caused serious obstruction to patient care pathways with subsequent sharp rises in waiting lists for consultation, RDT. Growing waiting lists translate into the human cost of prolonged symptoms, poor health, mental anguish and ultimately earlier death.[11] Most previous attempts to understand the RDT pathway have used linear queuing theory and there are few direct approaches using quantitative systems dynamics including feedback loops and complex flows.[12–14] This is important, because our model reveals the pathways between symptomatic, primary care and secondary and tertiary care to involve branched feed-forward and feedback loops. We fitted our model to open-access routinely collected data and created scenarios to illustrate how changes in capacities of constituent parts affect the overall flow. The results provide a stark illustration of the systemic nature of patient flow and that feedbacks within

the system can result in increased waiting lists despite seemingly increasing the number of appointments.

The modelling suggests that there is no rate-limiting step within the CVD care pathway that will reduce all waiting lists (and therefore reduce mortality). Indeed, our results suggest making a point change in the supply of treatments (eg, by increasing the number of intervention/operating theatre sessions) may well worsen flow through the system by creating a 'bottleneck' at the consultant appointment stage. The results also show that simply reducing all bottlenecks equally can have a less positive effect than a critical intervention at a single stage.

The problem of addressing postpandemic hospital waiting lists clearly applies more broadly than just to CVD in England. CVD was chosen because of its pre-eminence as a cause of death, and because treatment pathways are relatively self-contained within a single specialty. We chose to use England for this modelling because of the consistent availability of open-access routinely collected data. The specific conclusions based on these data do not necessarily apply to other conditions or medical systems, where the treatment pathways may be structured differently.

Systems dynamics is increasingly recognised as useful in healthcare modelling as it allows the qualitative components of causal diagrams which can be built from stakeholder discussion and existing knowledge with the quantitative modelling. In many cases, drawing the qualitative causal diagram is useful in its own right, as this can illustrate any unintended consequences of changes to the system—even without any numbers.[15] An advantage of a systems-dynamics approach over agent-based or machine learning models, is that it is a 'white-box' that enables straightforward assessment of the response to future shocks or interventions. A weakness of the approach is that it can oversimplify the effects of heterogeneities or small cohorts. The treatment of uncertainty in the model is at present uncomplicated, and more sophisticated stochastic simulation methods could be used to capture statistical ensembles. Another weakness is that the model does not model improved patient health as a result of treatments—some interventions may be more impactful at reducing death and improving health than others. We have also not explicitly modelled suppression in demand for secondary care. The NHS estimates 10 million fewer patients came forward to secondary care during the pandemic.[16] Some patients may have migrated to private treatment, some will have died of and with COVID-19 and some will have gained treatment for a CV condition via emergency care (eg, coronary disease manifesting as an acute coronary syndrome). The proportion of these is difficult to quantify; although emergency admissions in England for all causes in January 2022 are still reduced compared with January 2020.[16]

The study has a number of limitations which should be considered. The first is the model is a model—it may not completely account for all aspects of the system, including patient entry points, exit points, relationships

between components of the system, or other aspects. Another limitation is that the model only uses aggregated national data, and therefore, cannot be applied at the level of a particular region or hospital—limiting its use to large geographical area policy decisions. The model also does not account for the changes in patient health that occur because of the changes in the waiting list. CVD is progressive, and therefore, early intervention may have consequential effects later in the care pathway. This means that some interventions may be more impactful at improving health than others. The final limitation is that the observational nature of the data may limit the ability to accurately model future events due to the training of the model on historical data and the use of expectations for the future that do not fit prior experience. The model also requires key assumptions (eg, length of stay after an intervention) to hold across time periods within the model. Some assumptions, while reasonably chosen in the statistical analysis, may vary due to temporal changes in the environment, population needs or due to differences between distinct locations of the given services. An example of this is the rapidly reducing and varying length of stay after Transcatheter Aortic Valve Implant (TAVI).[17]

Further work needs to be conducted to evaluate use of our modelling approach in situ. Clearly the national-level aggregated data used here, suggest a use in policy setting. At a regional level, by fitting to local data sets, the model could also be used by care commissioners to decide on optimal resource allocation. Finally, the model could be used with appropriate data locally, by bed planners and clinical teams, to optimise individual care plans. Such studies are likely to reveal further constraints and feedback loops not currently captured in the model. For example, increasing consultant outpatient capacity could be implemented through overtime which would rely on already pressured staff and may lead to fewer staff being available. Such staffing constraints perhaps apply most acutely to nursing staff that have multiple functions within the pathway.

In summary, we have built a model of hospital waiting lists that has been fit to routinely collected data on the recovery of CV treatment after the COVID-19 pandemic. The study highlights an important new approach to modelling patient flow which, in contrast to black-box machine learning approaches, is transparent and contains tunable parameters that enable healthcare planners and policy-makers to rapidly test putative interventions and see their indirect effect on waiting lists for other parts of the system.

**Author affiliations**
[1]Department of Engineering Mathematics, University of Bristol, Bristol, UK
[2]Department of Mathematics, University of Southampton, Southampton, UK
[3]Department of Mathematics, University of Glasgow, Glasgow, UK
[4]British Heart Foundation, London, UK
[5]NHS Midlands and Lancashire Commissioning Support Unit, West Bromwich, UK
[6]Cardiac Surgery, University of Bristol, Bristol, UK
[7]Leeds Institute for Data Analytics, University of Leeds, Leeds, UK
[8]Leeds Institute of Cardiovascular and Metabolic Medicine, University of Leeds, Leeds, UK
[9]Cardiac Anaesthesia and Intensive Care, University of Bristol, Bristol, UK
[10]Department of Cardiac Anaesthesia, University Hospitals Bristol and Weston NHS Foundation Trust, Bristol, UK

**Contributors** SC collated and analysed the data, wrote and edited the manuscript. ARC designed the model, analysed and interpreted the data, wrote and edited the manuscript. RH designed the model and edited the manuscript. CC designed the model and edited the manuscript. JE designed the model. KC designed the model, interpreted the data and edited the manuscript. MW collated and analysed the data. GA interpreted the data and edited the manuscript. RN interpreted the data edited the manuscript. CG interpreted the data and edited the manuscript. BG designed the model, interpreted the data, wrote and edited the manuscript. BG is the overall guarantor of the manuscript.

**Funding** This study was funded by the British Heart Foundation, University Hospitals Bristol and Weston Charity and the NIHR Biomedical Research Centre at University Hospitals Bristol and Weston NHS Foundation Trust and the University of Bristol.

**Disclaimer** The views expressed are those of the authors and not necessarily those of the NIHR or the Department of Health and Social Care. The funders played no role in in study design; in the collection, analysis and interpretation of data; in the writing of the report; or in the decision to submit the paper for publication.

**Competing interests** None declared.

**Patient and public involvement** Patients and/or the public were not involved in the design, or conduct, or reporting, or dissemination plans of this research.

**Patient consent for publication** Not applicable.

**Provenance and peer review** Not commissioned; externally peer reviewed.

**Data availability statement** Data are available in a public, open access repository. No additional data are available. This study used open access, routinely collected data from the sources outlined in the main text.

**ORCID iDs**
Rebecca Hoyle http://orcid.org/0000-0002-1645-1071
Gianni Angelini http://orcid.org/0000-0002-1753-3730
Ramesh Nadarajah http://orcid.org/0000-0001-9895-9356
Ben Gibbison http://orcid.org/0000-0003-3635-6212

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

18  NHS Digital. Appointments in general practice. Available: https://digital.nhs.uk/data-and-information/ publications/statistical/appointments-in-general-practice [Accessed 01 Feb 2022].

19  NHS England. Monthly diagnostic waiting times and activity. Available: https://www.england.nhs.uk/ statistics/statistical-work-areas/diagnostics-waiting-times-and-activity/monthly-diagnostics-waiting-times-and-activity/ [Accessed 01 Feb 2022].

20  StatsWales. Patients registered at a GP practice. Available: https://statswales.gov.wales/Catalogue/Health-and-Social-Care/General-Medical-Services/patients-registered-at-a-gp-practice [Accessed 01 Feb 2022].

21  Office for National Statistics. Deaths registered in England and Wales — 21st century mortality. Available: https://www.ons.gov.uk/peoplepopulationandcommunity/birthsdeathsandmarriages/deaths/ datasets/the21stcenturymortalityfilesdeathsdataset/current [Accessed 01 Feb 2022].

22  NHS Digital. NHS sickness absence rates. Available: https://digital.nhs.uk/data-and-information/ publications/statistical/nhs-sickness-absence-rates [Accessed 01 Feb 2022].

