## [Reviewer comments · BMJ Open]

ARTICLE DETAILS

TITLE (PROVISIONAL)	Process modelling of NHS cardiovascular waiting lists in response to the COVID-19 pandemic
AUTHORS	Catsis, Salvador; Champneys, Alan R.; Hoyle, Rebecca; Currie, Christine; Enright, Jessica; Cheema, Katherine; Woodall, Mike; Angelini, Gianni; Nadarajah, Ramesh; Gale, Chris; Gibbison, Ben

VERSION 1 – REVIEW

REVIEWER	Paolo Landa Laval University, Operations and Decision Systems Department
REVIEW RETURNED	29-Jan-2023

GENERAL COMMENTS	In this manuscript is missing a detailed analysis of the literature review on both subject and methodology. It will be useful to include it in the manuscript in order to define the real challenges that previously were met by the researchers (e.g. some manuscripts of Sally Brailsford). The SD model is well defined and it clearly presented in terms of characteristics. However, I suggest that figure 1 might be enlarged or split in two main images as it will help the reader. Can you provide a detailed information on linear interpolation? Was the most appropriate? Can you provide some quality indicators? As it is a time series analysis, does present a seasonality or trend? The largest part of assumptions are justified. Why in the Monte Carlo simulation did you apply a 10% of variation? How you set this parameter? Did you add a value given from a distribution? Can you provide a sensitivity analysis with different values?
--

REVIEWER	Benjamin Horne Intermountain Medical Center, Intermountain Heart Institute
REVIEW RETURNED	07-Feb-2023

GENERAL COMMENTS	Catsis and colleagues from a multidisciplinary team across the UK evaluated the processes involved in healthcare delivery and choke points in those systems across the UK during the COVID-19 pandemic. A systems engineering approach was used to model the flow of patients into and across multiple services to establish a steady-state model using historical data and to examine changes to the model that may result in improved management of patients in the system. The mathematics are complex but generally well conducted. Major comments: 1. The major concerns with this paper are the limitations of the modeling. Some of these are described in summary on page 5, but are not included in a Limitations section in the Discussion. Other limitations exist. Certain limitations have been described briefly at
--

	some locations in the paper, including in the Methods, but those are dispersed and a summary of limitations would enhance the paper for readers. Please add some description of the following to a Limitations section before the final summary paragraph:  a. The model only uses aggregated national data (as per page 5 summary) b. The model does not account for the changes in patient health (as per page 5) c. The evaluation requires key assumptions to be made and to hold across time periods, which assumptions while reasonably chosen in the statistical analysis here may vary due to temporal changes in the environment, population needs, or etc., due to differences between distinct locations of the given services, or other changes or differences. d. The statistical model that was chosen may not completely account for all aspects of the system, including patient entry points, exist points, relationships between components of the system, or other aspects. e. The observational nature of the data may limit the ability to accurately model future events due to the training of the model on historical data and the use of expectations for the future that do not fit prior experience. 2. On page 8, line 58, it would be helpful to clarify the sentence, "Note that we used the StatsWales data only to correct England and Wales mortality data to England only." It isn't clear whether this means that the mortality data was standardized to English rates, if coding of causes of death was different in the Welsh data, or whether it means something else.
--	---

VERSION 1 – AUTHOR RESPONSE

Reviewer: 1

Dr. Paolo Landa, Laval University

Comments to the Author:

In this manuscript is missing a detailed analysis of the literature review on both subject and methodology. It will be useful to include it in the manuscript in order to define the real challenges that previously were met by the researchers (e.g. some manuscripts of Sally Brailsford).

It would be unusual in a manuscript published in BMJ Open to provide a detailed literature review as part of a novel manuscript. These are usually confined to one paragraph in the discussion – we have extended this slightly to include a short overview of systems dynamics modelling for the clinician.

The SD model is well defined and it clearly presented in terms of characteristics. However, I suggest that figure 1 might be enlarged of split in two main images as it will help the reader.

We will defer to the editorial team to decide on this point.

Can you provide a detailed information on linear interpolation? Was the most appropriate?

As stated in the supplementary materials, the time-series with the smallest collection time scale is the Appointments in General Practice series which is collected daily. In order to use the data to parameterise a Systems Dynamics model, each time-series is processed to produce a daily time-series. An overall time period over which to collect data to produce the model is chosen. For each data series, if it is collected daily we do not need to interpolate the data. However, if it is collected monthly or yearly, we need to interpolate the data to produce a daily time-series. We make the simplest assumption that the data in such a series follows a linear path between two consecutive data points whether that be weekly, monthly or yearly. Using this assumption, we use the method `interp1d` from the well documented Scipy package for Python to compute a one- dimensional linear

interpolation between the available data points in that series. This method returns a function which we can evaluate daily to produce a daily time-series.

Can you provide some quality indicators?

Care quality indicators are not available from publicly accessible data and so fall outside the scope of the data contained in our model.

As it is a time series analysis, does present a seasonality or trend?

There is a section in the manuscript in the results called Description of the Data which provides some analysis of the trends in the collected data. There are also graphs for each of the raw and processed time-series in the supplementary material which show the trends. It is difficult to provide an overall summary of the trend across the data (and hence we have not included it in the main manuscript) but it can be seen that most of the time-series follows the rough trend of:

1. Increasing or fluctuating about a mean value until the beginning of the 1st lockdown (March 2020).
2. Decreasing or remaining fairly consistent through the lockdown.
3. Increasing with a time-delay after the lockdown.

The largest part of assumptions are justified.

Why in the Monte Carlo simulation did you apply a 10% of variation? How you set this parameter? Did you add a value given from a distribution? Can you provide a sensitivity analysis with different values? As stated in section C.7 of the supplementary materials section, we implemented a Monte Carlo sampling procedure on one of the free parameters to illustrate how uncertainty in the model parameters may be propagated forward in the model predictions. We specifically chose to sample the free parameter Beta_cc representing the proportion of consultant appointments with referral to another consultant. We sampled this from a uniform distribution over the interval of +/- 5% of the value of Beta_cc calculated from the optimisation problem in the Post-lockdown period. We then ran some 100 simulations for different possible interventions as detailed in section C.7.2 with results shown in section D of the supplementary material. In these figures we show the projected mean and standard deviation of the different waiting list populations computed from the 100 simulations with different interventions. The values we chose to present were plausible illustrative examples (it is unlikely the NHS would be able to increase appointment supply or operating capacity by 50% for example). Five and 10% were plausible increases in the supply within the NHS.

Reviewer: 2

Dr. Benjamin Horne, Intermountain Medical Center, Stanford University

Comments to the Author:

Catsis and colleagues from a multidisciplinary team across the UK evaluated the processes involved in healthcare delivery and choke points in those systems across the UK during the COVID-19 pandemic. A systems engineering approach was used to model the flow of patients into and across multiple services to establish a steady-state model using historical data and to examine changes to the model that may result in improved management of patients in the system. The mathematics are complex but generally well conducted.

Major comments:

1. The major concerns with this paper are the limitations of the modeling. Some of these are described in summary on page 5, but are not included in a Limitations section in the Discussion. Other limitations exist. Certain limitations have been described briefly at some locations in the paper, including in the Methods, but those are dispersed and a summary of limitations would enhance the

paper for readers. Please add some description of the following to a Limitations section before the final summary paragraph:

- a. The model only uses aggregated national data (as per page 5 summary)
- b. The model does not account for the changes in patient health (as per page 5)
- c. The evaluation requires key assumptions to be made and to hold across time periods, which assumptions while reasonably chosen in the statistical analysis here may vary due to temporal changes in the environment, population needs, or etc., due to differences between distinct locations of the given services, or other changes or differences.
- d. The statistical model that was chosen may not completely account for all aspects of the system, including patient entry points, exist points, relationships between components of the system, or other aspects.
- e. The observational nature of the data may limit the ability to accurately model future events due to the training of the model on historical data and the use of expectations for the future that do not fit prior experience.

We have now included an expanded section within the discussion about the limitations of the model

2. On page 8, line 58, it would be helpful to clarify the sentence, "Note that we used the StatsWales data only to correct England and Wales mortality data to England only." It isn't clear whether this means that the mortality data was standardized to English rates, if coding of causes of death was different in the Welsh data, or whether it means something else.

We have now changed this sentence to clarify that StatsWales data was used to remove the Welsh data from the combined England and Wales data. For some data sources, only combined England and Wales data (rather than England alone) was available.

VERSION 2 – REVIEW

REVIEWER	Paolo Landa Laval University, Operations and Decision Systems Department
REVIEW RETURNED	05-Jun-2023

GENERAL COMMENTS	The authors have answered to my questions. By my side, the paper is ready for being accepted.
---

REVIEWER	Benjamin Horne Intermountain Medical Center, Intermountain Heart Institute
REVIEW RETURNED	10-Jun-2023

GENERAL COMMENTS	The authors have responded to all of my concerns
--